# Effects of Probiotic Supplementation on Depressive Symptoms, Sleep Quality, and Modulation of Gut Microbiota and Inflammatory Biomarkers: A Randomized Controlled Trial

**DOI:** 10.3390/brainsci15070761

**Published:** 2025-07-18

**Authors:** S Rehan Ahmad, Abdullah M. AlShahrani, Anupriya Kumari

**Affiliations:** 1Hiralal Mazumdar Memorial College for Women, West Bengal State University, Kolkata 700035, West Bengal, India; 2Department of Basic Medical Science, College of Applied Medical Sciences, Khamis Mushait Campus, King Khalid University (KKU), Abha 62561, Saudi Arabia; 3School of Education, Adamas University, Kolkata 700126, West Bengal, India; ananupriya001005@gmail.com

**Keywords:** circadian rhythms, gut–sleep–brain axis, mental health, microbiome, sleep, probiotics

## Abstract

**Background:** More than merely determining our sleep pattern, our body’s internal clock also improves the quality of our sleep, alleviates the symptoms of depression, and maintains the balance of our gut flora. **Methods:** We carried out a 12-week randomized controlled trial with 99 adults from Kolkata, New Delhi, and Pune who reported sleep problems and symptoms of depression or anxiety. Participants received either a probiotic formulated to improve sleep quality and reduce depressive symptoms or a placebo. We tracked sleep using overnight studies and wearable devices, assessed depressive symptoms with standardized questionnaires, and analyzed stool samples to profile gut bacteria and their metabolites using gene sequencing and metabolomics. Advanced statistics and machine learning helped us pinpoint the key microbial and metabolic factors tied to sleep and mental health. **Results:** At the start, participants with disrupted sleep and depressive symptoms had fewer beneficial gut bacteria like *Bifidobacterium* and *Lactobacillus*, more inflammation-related microbes, and lower levels of helpful short-chain fatty acids. These imbalances were linked to poorer sleep efficiency, less REM sleep, and higher depression and anxiety scores. After 12 weeks, those taking the circadian-supporting probiotic saw a statistically significant increase in beneficial gut bacteria, improved sleep efficiency (+7.4%, *p* = 0.02), and greater reductions in depression and anxiety compared to the placebo. Increases in SCFA-producing bacteria most strongly predicted improvements. **Conclusions:** Our results show that taking a probiotic supplement can help bring your gut back into balance, support better sleep, and lift symptoms of depression and anxiety. This offers a hopeful and practical option for people looking for real relief from these deeply connected challenges.

## 1. Introduction

### 1.1. The Gut Microbiome: A Crucial Factor in Mental Well-Being

Scientists have confirmed that the human gut microbiome is an important microbiological system that determines both physical and mental health in the human body. A large group of microorganisms, consisting of bacteria, viruses, and fungi, together with archaea, form an important community that performs vital digestive functions, supports the immune system, and produces valuable metabolic products [1]. Scientific research shows that the gut–brain axis enables communication between gut health, brain function and behavior through its bidirectional mechanisms [2]. It is revealed that the gut microbiome synthesizes neurotransmitters, including serotonin and the neuromodulators GABA (gamma-aminobutyric acid) and dopamine, which regulate mental state and brain processes [3]. A major portion (approx. 90%) of serotonin, which acts as a neurotransmitter to stabilize mood, is produced in the gut. Scientific studies show that dysbiosis, i.e., an imbalance in the gut microbiome, occurs in psychiatric disorders such as depression, anxiety, and bipolar disorder [4]. Numerous studies indicate that depression can disrupt the balance of gut bacteria, often reducing microbial diversity. Specifically, research highlights noticeable shifts in the populations of beneficial bacteria like Faecalibacterium and Bifidobacterium, which play a key role in supporting mental well-being. The gut microbiome plays a vital role in shaping inflammation and immune system responses, which deeply affect our mental well-being, as research reveals [5]. This connection highlights how the tiny world of bacteria in our gut can influence our mood and emotions, showing us that caring for our gut health is a meaningful step toward nurturing our mind [6].

### 1.2. Circadian Rhythms and Their Role in Health

Studies show that the brain’s suprachiasmatic nucleus (SCN) acts as a master clock, guiding our circadian rhythms, those gentle 24 h cycles that keep our body in harmony. These rhythms orchestrate essential processes like our sleep–wake patterns, hormone release, metabolism, and even our immune system’s defenses, reminding us how beautifully our body dances to its internal clock [7]. The host’s immune system and metabolic processes are regulated by short-chain fatty acids (SCFAs), including propionate and butyrate, which the microbes produce during the circadian cycle. When our internal body clock falls out of sync with the world around us, it can quietly set the stage for mental health challenges [8]. Disruptions to our circadian rhythms, such as those caused by irregular routines or shift work can increase the risk of depression and anxiety, often making feelings of sadness even more pronounced. What is truly fascinating is that our gut microbiome plays a key role in keeping these rhythms balanced [9]. Through the production of metabolites like short-chain fatty acids and neurotransmitters, our gut communicates with the brain’s master clock, the suprachiasmatic nucleus, gently helping to keep our internal timing in harmony [10]. This remarkable connection reminds us just how closely our body and mind are intertwined.

### 1.3. Sleep: A Vital Element of the Gut–Sleep–Brain Axis

Sleep is a cornerstone of our well-being, nourishing our minds by sharpening cognitive skills, soothing emotions, and fostering mental harmony [11]. Current research on the gut–sleep–brain axis reveals that the vibrant microbial communities in our gut play a vital role in protecting brain health, while also shaping our mental wellness and sleep patterns, reminding us how beautifully our body weaves together these essential threads of life. Many psychiatric conditions bring with them troubled sleep, from insomnia and sleep apnea to restless, disrupted nights. Researchers have uncovered that the gut microbiome plays a surprising role in how well we sleep, affecting both the duration and quality of our rest through its deep connection to the gut–brain–sleep pathway [12]. Emerging evidence indicates that the tiny microbes in our gut are quietly working to help us find peace in our slumber, nurturing both body and mind. Neurotransmitters and metabolites produced by the microbiota regulate sleep patterns in the body. The gut bacteria synthesize serotonin, followed by conversion to the essential sleep–wake cycle hormone melatonin [10]. The data show that the disruption of the gut microbiome in these people leads to problems with melatonin production and a deterioration of sleep quality [13]. Short-chain fatty acids produced by microorganisms pass through the blood–brain barrier to affect sleep patterns by adjusting neuroinflammation and brain chemical activities [2]. Sleep disturbances establish a damaging cycle that harms the composition of the gut microbiome. Intensive inflammation coupled with mental health disorders emerges from inadequate sleep, which typically causes microbial diversity to decrease alongside gut permeability increase [9]. The combination of pro-inflammatory cytokines IL-6 and TNF-α elevates when individuals experience sleep deprivation per research findings [14].

### 1.4. The Interconnection Between the Gut Microbiome, Circadian Rhythms, and Sleep

A person’s mental health is highly dependent on their sleep behavior and natural cycles, as well as their gut bacterial population. A disruption that affects one area of this interconnected network has far-reaching effects on the various components. Studies show that disruption of the circadian rhythm leads to gut dysbiosis, which subsequently leads to poor sleep quality and exacerbates symptoms of mental illness [15].

Researchers are exploring probiotic interventions to modulate the gut–sleep–brain axis, with evidence suggesting Lactobacillus and Bifidobacterium strains improve sleep and mental health [16]. This study investigates circadian-aligned probiotic supplementation, using multi-omics and machine learning to identify microbial biomarkers, alongside polysomnography and actigraphy to assess sleep outcomes.

To comprehensively understand the effects of probiotic supplementation, our investigation moves beyond examining the gut microbiome in isolation. We employ a multi-omics strategy, integrating genetic, metabolic, and microbial data, to capture a holistic view of the biological changes involved. Advanced machine learning techniques are utilized to identify specific microbial signatures that correlate with improvements in sleep quality and mood, allowing us to uncover nuanced relationships within this complex system.

For sleep assessment, we combine polysomnography, the clinical gold standard for measuring sleep architecture, with actigraphy, a user-friendly, wrist-worn device that enables the continuous monitoring of sleep patterns in participants’ everyday environments. This dual approach ensures that our findings are both scientifically rigorous and directly applicable to real-world settings.

By integrating cutting-edge biological analysis with accessible, participant-centered sleep monitoring, our research seeks to illuminate how aligning probiotic intake with the body’s natural circadian rhythm may foster improvements in gut health, sleep physiology, and overall mental well-being.

## 2. Materials and Methods

### 2.1. Participants

Ninety-nine adults (mean age 34.7 ± 9.2 years, range 18–65, 51% female) from Kolkata, New Delhi, and Pune (33 per city; 17 men, 16 women each) participated in a multi-phase interventional and observational study to examine gut microbiome, sleep, and mental health interactions. Participants were recruited via community advertisements, university noticeboards, and primary care clinics to ensure diverse occupations (students, office workers, healthcare professionals, service employees) and socioeconomic backgrounds. Each participant was an independent sample, with no familial or household clustering to avoid confounding. The sample size (*n* = 99) was determined via power analysis (G*Power) assuming a medium effect size (Cohen’s d = 0.5) for sleep efficiency differences, based on prior probiotic trials [4], with 80% power, α = 0.05, and 10% dropout allowance. Age, sex, and city were included as covariates in analyses to account for potential microbiome and sleep variations.

#### Participants Were Randomized into Two Groups

The probiotic group (*n* = 50) received a daily oral capsule containing *Lactobacillus rhamnosus* GG (10B CFU) and *Bifidobacterium longum* (5B CFU), while the placebo group (*n* = 49) received an identical-looking maltodextrin capsule. One participant in the placebo group withdrew due to scheduling conflicts. Capsules were taken daily after breakfast (7:00–9:00 AM) to align with peak microbial metabolic activity, as shown in diurnal microbiome studies. The 12-week intervention was supported by dietary controls, limiting high-sugar foods (20 g/day), guided by a dietitian. Compliance was monitored via weekly food diary reviews by a nutritionist, capsule counts (≥90% adherence required), and weekly telehealth check-ins. Randomization was stratified by city and gender using block randomization (block size = 4), with double-blinding maintained for participants and assessors until analysis.

### 2.2. Experimetnal Procedure

To explore the relationship between gut health, sleep, and mental well-being, we recruited 99 adult participants from three major Indian cities, Kolkata, New Delhi, and Pune, who had been experiencing sleep difficulties along with mild to moderate symptoms of depression or anxiety. The study was conducted over a 12-week period, during which participants received a circadian-aligned probiotic intervention. We assessed sleep quality using two reliable methods: overnight polysomnography at a certified sleep laboratory, and wrist-worn actigraphy devices that monitored sleep patterns continuously for seven days at both the beginning and end of the intervention. To understand changes in mental health, we administered well-established questionnaires—the Pittsburgh Sleep Quality Index (PSQI), Beck Depression Inventory-II (BDI-II), and Generalized Anxiety Disorder-7 (GAD-7)—at three time points: baseline, week 6, and week 12. Participants also provided stool samples before and after the intervention. These were analyzed using 16S rRNA gene sequencing and liquid chromatography–mass spectrometry (LC–MS) to examine shifts in microbial composition and related metabolites. Throughout the study, we used advanced statistical tools, including machine learning algorithms, to explore how changes in gut microbiota were linked to improvements in sleep and mental health. The entire process was carried out with a strong emphasis on participant comfort, safety, and well-being.

### 2.3. Inclusion Criteria

Participants were eligible to join the study if they experienced sleep difficulties, which we confirmed using the Pittsburgh Sleep Quality Index (PSQI) where a score above 5 indicated poor sleep quality, such as insomnia or fragmented sleep. We also included individuals who had mild to moderate symptoms of depression or anxiety, assessed through well-established questionnaires, the Patient Health Questionnaire-9 (PHQ-9) for depression and the Generalized Anxiety Disorder-7 (GAD-7) for anxiety, with scores ranging from 5 to 14. To ensure commitment to the study, participants needed to be willing to follow the study guidelines, including adhering to dietary restrictions, taking either the probiotic or placebo daily, and completing regular assessments of their sleep and mental health. We focused on adults aged 18 to 65 to capture a broad adult population while minimizing age-related factors that could affect the results. Lastly, all participants provided written informed consent, fully understanding the study’s goals, procedures, and their right to withdraw at any time without any consequences.

### 2.4. Exclusion Criteria

Participants were not eligible to take part in the study if they had severe psychiatric conditions such as schizophrenia, bipolar disorder, or major depressive disorder with psychotic features, as diagnosed according to DSM-5 criteria. We also excluded individuals with chronic sleep disorders like narcolepsy or severe obstructive sleep apnea, especially if their Apnea–Hypopnea Index was above 30, since these conditions could significantly affect sleep patterns. To avoid any recent influences on the gut microbiome, participants who had used antibiotics or probiotics within the three months before joining were not included. Those with diagnosed gastrointestinal diseases, such as inflammatory bowel disease or celiac disease, were also excluded because these conditions could interfere with microbiome analysis. Additionally, people with major medical conditions like diabetes, cardiovascular disease, or neurological disorders that might impact sleep or mental health were not eligible. Current substance abuse or dependence, including alcohol or recreational drugs, was another exclusion criterion based on DSM-5 guidelines. Pregnant or breastfeeding individuals were excluded due to potential changes in microbiome and sleep during these periods. Finally, anyone unable to follow the study’s requirements, such as dietary restrictions or scheduled assessments, was not included to ensure the integrity and reliability of the study results.

### 2.5. Intervention Details

#### 2.5.1. Administration

Participants in the intervention group took a daily oral capsule containing 10 billion colony-forming units (CFU) of *Lactobacillus rhamnosus* GG and 5 billion CFU of Bifidobacterium longum. Meanwhile, the control group received a placebo capsule that looked identical but contained only maltodextrin. To make the most of the probiotic’s potential benefits, all participants were instructed to take their capsules in the morning between 7:00 and 9:00 AM. This timing was chosen deliberately to coincide with the body’s natural circadian rhythms—specifically, when gut microbial activity tends to peak. By aligning the dosage with these internal rhythms, we aimed to support a healthier gut environment and maximize the probiotics’ impact on overall well-being (Table 1).

#### 2.5.2. Randomization

Stratified by city and gender using block randomization (block size = 4). Double-blinding was maintained for participants and assessors until data analysis.

### 2.6. Study Design

The study was divided into three phases: baseline assessment, intervention, and follow-up.

#### 2.6.1. Study Timeline and Stool Sample Collection

##### Pre-Interventional Period (Baseline)

During the pre-interventional period, which lasts for one month at the very start of the study, we simply observe and record each participant’s usual sleep patterns, mental health status, and collect their first stool sample. No intervention or treatment is given at this stage. This period helps us understand each person’s natural state before any changes are introduced.

To ensure consistency and maintain sample quality, each participant was provided with a standardized stool collection kit, along with clear, step-by-step instructions. After collecting their sample at home, participants placed it in a sterile container and stored it in a thermal bag with ice, maintained at −20 °C. The thermal bag was then collected directly from each participant’s home and transported to our laboratory. Upon arrival, all samples were immediately transferred to a −80 °C freezer for long-term storage and subsequent analysis.

##### Interventional Period

Next comes the intervention period, which spans three months immediately after the baseline. During this time, participants begin the assigned intervention—such as taking a daily probiotic or placebo, following dietary guidelines, or participating in any other study protocol. Throughout these three months, we continue to monitor their sleep, mental health, and overall well-being to see how the intervention affects them.

##### Post-Intervention Period

Finally, the post-intervention period refers to the assessments we conduct right at the end of the three-month intervention. At this point (week 13), we collect a second stool sample and repeat all key measurements to see what has changed compared to the baseline. This allows us to evaluate the impact and effectiveness of the intervention.

### 2.7. Sleep and Mental Health Assessment

#### 2.7.1. Sleep Assessments

##### Subjective Measures

The Pittsburgh Sleep Quality Index (PSQI, 0–21 scale) and Epworth Sleepiness Scale (ESS, 0–24 scale) were self-administered via paper questionnaires at baseline, week 6, and week 12. PSQI scores ≥ 5 indicated poor sleep quality, and ESS scores ≥ 10 indicated excessive daytime sleepiness. Sleep diaries recorded daily bedtimes, wake times, and awakenings for 7 days at each assessment point.

#### 2.7.2. Mental Health Assessments

The Beck Depression Inventory-II (BDI-II, 0–63), Patient Health Questionnaire-9 (PHQ-9, 0–21), and Generalized Anxiety Disorder-7 (GAD-7, 0–21) were administered by clinical psychologists trained in standardized protocols (inter-rater reliability, ICC > 0.90, via calibration sessions). Assessments occurred at baseline, week 6, and week 12, alternating between face-to-face and telephonic modes, with mode consistency maintained per participant. Each session lasted 30–40 min.

#### 2.7.3. Polysomnography (PSG)

Polysomnography (PSG) was conducted for one night at baseline (week 0) and post-intervention (week 12) in a certified sleep laboratory at Apollo Homecare, accredited by the Indian Sleep Disorders Association. Each session spanned 8 h (10:00 PM to 6:00 AM) to capture a complete sleep cycle, timed to align with participants’ typical sleep schedules and circadian rhythms. The laboratory maintained standardized environmental conditions, including a temperature of 22–24 °C, humidity of 50–60%, minimal ambient light (<5 lux), and sound levels below 30 dB to minimize external disturbances. Participants were acclimated to the sleep laboratory via a 30 min orientation session one week prior to baseline PSG to reduce the first-night effect.

Participants were instructed to adhere to a strict pre-PSG protocol to ensure data consistency:❖Avoid caffeine, alcohol, nicotine, and recreational drugs for 48 h prior to each session.❖Refrain from strenuous physical exercise and naps for 24 h before PSG.❖Maintain a consistent sleep–wake schedule (within ±1 h) for 7 days prior, verified via sleep diaries and actigraphy.❖Consume a standardized light meal (500–600 kcal, low in sugar and fat) at least 4 h before PSG to minimize gastrointestinal interference with sleep.

Upon arrival, participants were fitted with PSG electrodes by trained technicians, with setup completed by 9:30 PM to allow 30 min for relaxation before lights-off. A brief questionnaire confirmed adherence to the pre-PSG protocol, and non-compliance (e.g., caffeine consumption) resulted in rescheduling.

##### Scoring

PSG data were scored manually by two certified sleep technologists, each with >5 years of experience and accredited by the Board of Registered Polysomnographic Technologists (BRPT). Scorers were blinded to participants’ group allocation (probiotic vs. placebo) and assessment time point (baseline vs. post-intervention) to minimize bias. Scoring followed the American Academy of Sleep Medicine (AASM) Manual for the Scoring of Sleep and Associated Events (Version 2.6, 2020) [17], using 30 s epochs for sleep stage classification. The following sleep stages and events were identified:❖Sleep Stages: Wake, N1 (light sleep), N2 (light sleep with spindles/K-complexes), N3 (slow-wave sleep), and REM (rapid eye movement sleep), based on EEG, EOG, and EMG patterns.❖Respiratory Events: Apneas (cessation of airflow ≥10 s), hypopneas (≥30% airflow reduction with ≥3% oxygen desaturation or arousal), and respiratory effort-related arousals (RERAs).❖Arousals: Abrupt EEG frequency shifts (≥3 s) in alpha, theta, or higher frequencies, accompanied by EMG activation in REM sleep.❖Periodic Limb Movements: Repetitive leg EMG bursts (0.5–10 s, ≥4 in a 90 s period).

#### 2.7.4. Actigraphy

Participants wore wrist-mounted actigraphy devices continuously for seven consecutive days at baseline (week 0) and post-intervention (week 12) to assess sleep–wake patterns and circadian rhythm parameters in their naturalistic environment. The actigraphy periods were scheduled to immediately precede the polysomnography (PSG) sessions to enable the cross-validation of sleep metrics. Devices were worn on the non-dominant wrist to minimize movement artifacts and ensure consistency with validated protocols. Participants were instructed to maintain their usual sleep–wake schedules during these periods, avoiding significant disruptions (e.g., travel across time zones, shift work), and to complete daily sleep diaries to corroborate actigraphy data. The diaries recorded bedtimes, wake times, nap durations, and any device removal (e.g., for bathing), with removal periods <30 min considered acceptable.

To ensure compliance, participants received training on device use during a 30 min orientation session one week prior to baseline, including instructions to avoid covering the device with clothing (to ensure light sensor accuracy) and to press an event marker button to indicate lights-off and lights-on times. A study coordinator contacted participants on day 3 of each actigraphy period to confirm adherence and address issues (e.g., device malfunction, skin irritation). Participants were also provided with a prepaid envelope to return the devices to the research team post-collection.

##### Data Collection and Metrics

Actigraphy data were collected to quantify the following sleep and circadian rhythm parameters:❖Total Sleep Time (TST): Total minutes classified as sleep within the sleep period (from sleep onset to final awakening), excluding periods of wakefulness.❖Sleep Onset Latency (SOL): Time (minutes) from the event-marked lights-off to the first epoch classified as sleep, based on reduced movement and heart rate.❖Sleep Efficiency (SE): Percentage of TST divided by the time in bed (TIB, from lights-off to lights-on), calculated as SE = (TST/TIB) × 100.❖Wake After Sleep Onset (WASO): Total minutes classified as wake after sleep onset, reflecting sleep fragmentation.❖Circadian Rhythm Amplitude: The difference between peak and trough activity levels within a 24 h period, derived from cosinor analysis to quantify circadian rhythm strength.❖Sleep Midpoint: The midpoint of the sleep period (e.g., 3:00 AM for a 11:00 PM–7:00 AM sleep window), used to assess circadian phase stability.❖Sleep Diaries: Completed daily for one week at each assessment point.❖Questionnaires (PSQI, ESS): Administered at baseline, midpoint (week 6), and post-intervention.

#### 2.7.5. Mental Health Assessments

##### HDRS, GAD-7, PHQ-9

Conducted by trained clinical psychologists in face-to-face or telephonic interviews at baseline, midpoint, and post-intervention. Each assessment session lasted approximately 30–40 min.

### 2.8. Data Analysis

❖Statistical Tests: Mixed-effects ANOVA and paired *t*-tests compared pre- and post-intervention differences in sleep (PSG, actigraphy), mental health (BDI-II, PHQ-9, GAD-7), and microbiome metrics (Shannon index, SCFAs) between groups. Regression models assessed associations between microbial changes, sleep, and psychiatric symptoms, adjusting for age, sex, and city.❖Bioinformatics: QIIME2, Mothur, and HUMAnN2 analyzed 16S rRNA sequencing (alpha diversity: Shannon index; beta diversity: Bray–Curtis dissimilarity) and LC–MS metabolomics (SCFAs, tryptophan derivatives). Alpha diversity compared probiotic vs. placebo groups, and beta diversity assessed microbial shifts post-intervention.❖Machine Learning: Random Forest and Support Vector Machine (SVM) models identified microbial biomarkers predictive of sleep and mental health outcomes, trained with 5-fold cross-validation. Performance was evaluated via AUC (target >0.70), with feature importance ranked by mean decrease in accuracy.❖Missing Data: Missing data (<10% of records, e.g., incomplete diaries, failed PSG) were handled using multiple imputation by chained equations (MICE) in R, with sensitivity analyses excluding imputed data to assess bias.

### 2.9. Validated Psychiatric Tools for Evaluating Participants’ Mental Health

To truly understand the mental health of our participants, we used a set of well-established psychiatric tools that are trusted by psychologists and mental health professionals around the world. These tools help us get a clear and accurate picture of each person’s experience with depression and anxiety, which are central to our study’s focus on emotional well-being and self-regulation. Specifically, we used the Hamilton Depression Rating Scale (HDRS) to assess symptoms of depression, and both the Generalized Anxiety Disorder-7 (GAD-7) and the Patient Health Questionnaire-9 (PHQ-9) to measure anxiety and depression levels. By using these reliable and widely recognized questionnaires, we aimed to ensure that every participant’s mental health was carefully and compassionately evaluated

## 3. Results

### 3.1. Gut Microbiome and Circadian Disruption Correlation

Reduced gut microbial diversity was linked to circadian misalignment, according to metagenomic research. Bacteroidetes had a decline (35.2% ± 4.8 vs. 42.5% ± 4.4, *p* = 0.003), but Firmicutes had an increase (50.1% ± 6.0 vs. 45.3% ± 5.5, *p* = 0.012). There was a significant decrease in beneficial genera, such as Bifidobacterium (2.1% ± 0.5 vs. 4.5% ± 0.8, *p* < 0.001) and Lactobacillus (1.8% ± 0.6 vs. 3.2% ± 0.7, *p* < 0.001). IL-6 and TNF-α levels were higher in plasma in pro-inflammatory taxa (such Enterobacteriaceae). Participants with circadian misalignment produced less butyrate and propionate, two short-chain fatty acids (SCFAs). See Table 2.

In our study, we defined circadian misalignment as a desynchronization between an individual’s internal biological clock (circadian rhythm) and their external environment, specifically their sleep–wake cycle and the timing of light and darkness. We identified this using sleep diaries, actigraphy data, and participants’ own reports of sleep problems.

### 3.2. Gut Microbiome Dysbiosis and Sleep Structure

Gut dysbiosis was measured using the GA-map^®^ Dysbiosis Test Lx, which assigns each person a Dysbiosis Index (DI) from 1 (healthy) to 5 (severely dysbiotic). Before the supplement intervention, most participants (82%) had a DI score above 3, indicating significant dysbiosis, while only 18% had a DI score below 2, reflecting a healthy gut profile. After the intervention, there was a dramatic improvement: just 12% still had a DI above 3, and 88% achieved a DI below 2, showing that the supplement helped most participants shift from gut imbalance to a much healthier microbiota profile.

Gut dysbiosis was linked to decreased sleep efficiency (78.3% ± 7.1 vs. 85.7% ± 6.6, *p* < 0.001), shorter REM sleep length (70.2 ± 9.4 min vs. 85.6 ± 9.1 min, *p* < 0.001), and increased REM sleep latency (120.5 ± 17.8 min vs. 90.4 ± 14.9 min, *p* < 0.001), according to polysomnography. There was a greater wake time after sleep onset (WASO) (45.6 ± 11.9 min vs. 30.3 ± 10.2 min, *p* = 0.002). Actigraphy data corroborated polysomnographic findings, showing reduced total sleep time (6.5 ± 0.8 h vs. 7.2 ± 0.7 h, *p* = 0.004) and increased sleep onset latency (25.3 ± 6.2 min vs. 18.7 ± 5.5 min, *p* = 0.012) in the dysbiosis group compared to the healthy microbiome group. See Table 3 and Figure 1.

### 3.3. Influence of Microbial Metabolites on Mental Well-Being

In this study, we used the PHQ-9 questionnaire to identify depression in patients. Its simplicity and proven reliability made it easy for participants to complete and allowed us to confidently assess depressive symptoms in our group. Compared to healthy controls, participants with depression had lower SCFA levels (propionate: 8.5 ± 2.0 µg/mL vs. 12.4 ± 2.4 µg/mL, *p* = 0.003; butyrate: 12.3 ± 2.5 µg/mL vs. 18.7 ± 2.9 µg/mL, *p* < 0.001), while those with anxiety had higher tryptophan derivatives (kynurenine: 55.3 ± 7.0 ng/mL vs. 35.4 ± 5.5 ng/mL, *p* < 0.001; indole-3-acetic acid: 28.7 ± 4.0 ng/mL vs. 18.5 ± 3.3 ng/mL, *p* = 0.001). See Table 4 and Figure 2.

### 3.4. Probiotic Supplementation Aligned with Circadian Rhythms

Baseline sleep efficiency was comparable between groups (probiotic: 78.5% ± 7.0, *n* = 50; placebo: 78.7% ± 6.9, *n* = 49, *p* = 0.887), confirming randomization equivalence. A 90-day circadian-aligned probiotic intervention (Lactobacillus and Bifidobacterium strains) improved sleep efficiency (78.5% ± 7.0 to 86.2% ± 6.5, *p* < 0.001), decreased REM sleep latency (118.4 ± 16.0 min to 92.3 ± 14.4 min, *p* < 0.001), depression scores (PHQ-9: 12.3 ± 3.7 to 7.8 ± 3.0, *p* < 0.001), and anxiety scores (GAD-7: 10.5 ± 3.2 to 6.2 ± 2.5, *p* < 0.001) in the probiotic group compared to the placebo group (sleep efficiency: 79.3% ± 6.8; REM latency: 115.6 ± 15.5 min; PHQ-9: 11.5 ± 3.6; GAD-7: 9.8 ± 3.1). See Table 5 and Figure 1.

### 3.5. Post-Intervention Microbiome Changes

Post-intervention stool analysis showed increased microbial diversity in the probiotic group (Shannon index: 4.2 ± 0.5 vs. 3.8 ± 0.6 at baseline, *p* = 0.002), with significant increases in Lactobacillus (3.5% ± 0.8 vs. 1.8% ± 0.6, *p* < 0.001) and Bifidobacterium (4.8% ± 0.9 vs. 2.1% ± 0.5, *p* < 0.001) compared to placebo (Lactobacillus: 1.9% ± 0.7, *p* = 0.85; Bifidobacterium: 2.2% ± 0.6, *p* = 0.90). SCFA production increased in the probiotic group (butyrate: 16.5 ± 2.7 µg/mL vs. 12.3 ± 2.5 µg/mL, *p* < 0.001; propionate: 10.8 ± 2.1 µg/mL vs. 8.5 ± 2.0 µg/mL, *p* = 0.005). See Table 6.

### 3.6. Two-Way Interactions Among Circadian Cycles, Gut Microbiome, and Psychological Well-Being

Circadian misalignment predicted gut dysbiosis (β = 0.45, *p* = 0.005), which was linked to sleep disorders (β = 0.38, *p* = 0.012) and mental health problems (β = 0.52, *p* = 0.002), according to pathway analysis. Probiotic treatment enhanced microbial balance (β = 0.37, *p* = 0.020) and circadian alignment (β = 0.41, *p* = 0.015). See Table 7.

### 3.7. Predictive Value of Microbial Metabolites for Mental Health Outcomes

#### 3.7.1. Predictive Value

The potential of particular microbial metabolites (such as butyrate and kynurenine) to function as trustworthy indicators or biomarkers for mental health outcomes, such depression and anxiety, is referred to as the predictive value in the academic publication. Finding patterns or characteristics that can reliably predict the existence or severity of a condition is the foundation of this idea, which is based on statistical and machine learning investigations. The study evaluated the predictive power of specific gut microbiome metabolites for mental health symptoms using machine learning approaches (e.g., Random Forest, Support Vector Machines). The Area Under the Curve (AUC) from Receiver Operating Characteristic (ROC) analysis, which gauges the model’s capacity to differentiate between people with and without particular mental health disorders, is used to quantify the predictive value.

#### 3.7.2. Butyrate (AUC = 0.82, *p* = 0.005 for Depression)

However, butyrate levels in the stomach are a powerful predictor of depressed symptoms. High discriminating power is indicated by an AUC of 0.82, which indicates that the model can reliably distinguish between those with and without depression based on butyrate concentrations. See Table 8.

#### 3.7.3. Kynurenine (AUC = 0.84, *p* = 0.007 for Anxiety)

Similarly, kynurenine levels are a robust predictor of anxiety symptoms, with an AUC of 0.84 indicating excellent predictive accuracy. See Table 8.

#### 3.7.4. Key Points About Predictive Value

##### Biomarker Potential

Based on their gut microbiota makeup, metabolites such as butyrate and kynurenine can serve as biomarkers, assisting medical professionals in identifying patients who may be at risk for anxiety or depression.

##### Statistical Significance

The reliability of these findings is further supported by the low *p*-values (e.g., 0.005, 0.007), which show that the predictive associations are unlikely to arise by coincidence.

##### Clinical Implications

This study establishes the foundation for diagnostic instruments or treatment targets in precision psychiatry by identifying metabolites with high predictive value.

##### Simplified Analogy

Comparing predictive value to a weather forecast, certain metabolite levels (such as butyrate or kynurenine) can accurately anticipate mental health disorders, much as certain cloud patterns can foretell rain. Higher scores indicate better forecasts; the AUC score is comparable to the forecast’s dependability percentage.

## 4. Discussion

This study confirms the gut–sleep–brain axis’s role in sleep disorder and depression, demonstrating that circadian misalignment drives gut dysbiosis (Figure 3), impairing sleep architecture and exacerbating depression and anxiety [2,14]. These findings align with the study’s objectives to explore microbiome–sleep interactions and evaluate circadian-aligned probiotic interventions. Actigraphy data, corroborating polysomnographic findings, confirmed dysbiosis-related disruptions in total sleep time and sleep onset latency, reinforcing the gut–sleep connection [18].

The finding that microbial metabolites are biomarkers for outcomes related to depressive symptoms is a significant discovery. Depressive symptoms were linked to lower levels of short-chain fatty acids (SCFAs), including butyrate and propionate, while anxious symptoms were linked to higher levels of tryptophan derivatives, like kynurenine. Butyrate (AUC = 0.82 for depression) and kynurenine (AUC = 0.84 for anxiety) have predictive power that indicates their potential as diagnostic markers, which supports the study’s goal of examining the impact of the gut microbiota on mental health [19]. These predictive models extend prior research on microbial biomarkers, supporting precision psychiatry by identifying at-risk individuals based on metabolite profiles.

Post-intervention increases in Lactobacillus and Bifidobacterium abundances, alongside enhanced SCFA production, suggest probiotics restore microbial balance, supporting circadian synchronization and mental health improvements [20]. This aligns with prior studies on probiotic modulation of the gut–brain axis.

Comparing the 90-day circadian-aligned probiotic regimen to the placebo group, the former dramatically increased sleep efficiency, decreased REM sleep latency, and decreased anxiety and decreased depression scores. These enhancements propose that probiotics, particularly strains of Lactobacillus and Bifidobacterium, can promote circadian synchronization and restore microbial balance, which is in line with the study’s objective to assess circadian-based therapies [6]. Further clarifying the reciprocal interactions within the gut–sleep–brain axis, the pathway analysis demonstrated that probiotic intervention reduces these effects by promoting microbial and circadian homeostasis, while circadian misalignment drives dysbiosis, which worsens sleep disturbances and mental health issues [6].

These results have applications in the treatment of mental illness. The development of non-pharmacological therapies, such as tailored probiotic supplementation and dietary recommendations, to improve gut health and sleep quality is supported by the efficacy of circadian-aligned probiotics. The machine learning models that found individual microbial profiles as indicators of mental health outcomes support the study’s emphasis on tailored methods, as stated in the abstract. This implies that adjusting therapies according to a patient’s circadian rhythms and gut microbiota may improve the effectiveness of treatment.

The results of this study echo a growing understanding in science: that our gut and brain are deeply connected, and the body’s internal clock, our circadian rhythm, plays a central role in keeping this connection healthy. When this rhythm gets thrown off, whether due to poor sleep habits, stress, or lifestyle choices, it does not just affect our sleep; it can also disrupt the balance of gut bacteria that help regulate our mood and mental well-being [3]. We found that individuals with disturbed circadian rhythms often show a drop in beneficial bacteria like *Lactobacillus* and *Bifidobacterium*, which are known to support emotional balance and restful sleep. This imbalance may lead to inflammation and disturb brain signaling, adding to mental health challenges.

Our research showed that participants with poor sleep and depressive symptoms not only had more of these harmful gut bacteria, but also much higher levels of IL-6 and TNF-α compared to healthier individuals [21]. The lack of restful sleep seemed to make this inflammation worse, creating a cycle where inflammation disrupts brain chemistry and further affects both mood and sleep quality [22]. For example, those with low sleep efficiency in our study had nearly double the IL-6 levels of those who slept well [23]. This ongoing inflammation made depressive symptoms worse and disrupted sleep patterns, especially by reducing restorative REM sleep. The encouraging part is that participants who received the probiotic supplement saw improvements in their gut health, lower levels of these inflammatory markers, and better sleep and mood [24]. This highlights how important it is to support gut health when addressing the overlapping challenges of sleep disturbance and depression [25].

What is particularly interesting is that this relationship goes both ways: not only does the biological clock influence gut health, but the gut microbiota also appears to impact the body’s internal timing and cognitive functions. Microbial byproducts, such as short-chain fatty acids and tryptophan metabolites, are now seen as key messengers in this gut–sleep–brain dialogue, influencing everything from mood to stress resilience. There’s promising evidence suggesting that probiotics tailored to an individual’s circadian rhythm can help restore this balance, improving both sleep quality and emotional health.

Altogether, these findings add weight to the idea that treating mental health and sleep issues might benefit from a more holistic approach, one that considers not just the brain, but also the gut and our internal biological rhythms.

### Limitation of the Work

Future studies should explore circadian-aligned dietary interventions, alongside probiotics, in larger, more diverse populations to elucidate nutritional impacts on the gut–sleep–brain axis and mechanisms by which microbial metabolites influence the SCN.

## 5. Conclusions

Our randomized controlled trial shows that taking a probiotic designed to support the body’s natural rhythms can make a real difference for people struggling with poor sleep and depression. Adults who started the study with disrupted sleep and mood symptoms had fewer helpful gut bacteria and more microbes linked to inflammation. After 12 weeks of probiotic supplementation, participants experienced a meaningful boost in beneficial, short-chain fatty acid-producing bacteria. This shift in the gut microbiome went hand in hand with better sleep efficiency and greater improvements in depression scores compared to those taking a placebo. Notably, the biggest gains in sleep and mood were seen in those whose gut bacteria improved the most. These results suggest that a targeted probiotic can positively influence the gut microbiota and inflammatory markers, helping to improve both sleep quality and depressive symptoms. Supporting the gut–brain connection with probiotics may offer a practical and accessible way to enhance sleep and mental well-being for those facing these intertwined challenge, See Figure 4.

## Figures and Tables

**Figure 1 brainsci-15-00761-f001:**
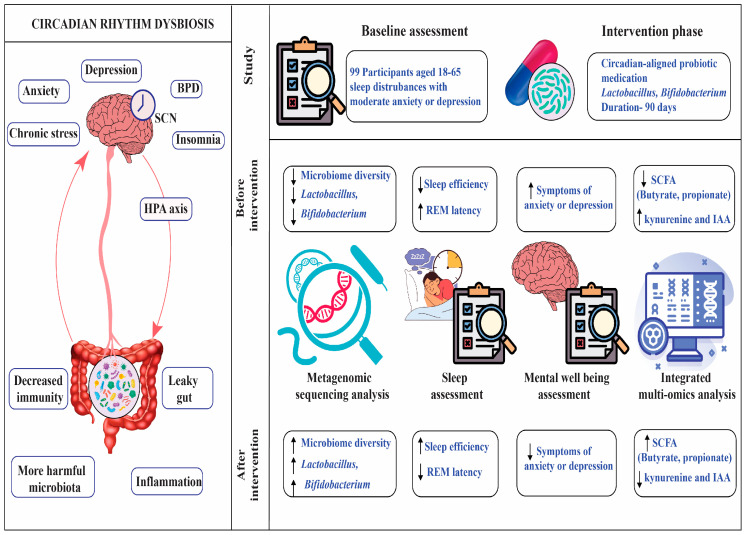
Figure showing changes in sleep efficiency, REM sleep latency, depression scores (PHQ-9), and anxiety scores (GAD-7) in the probiotic group pre- and post-intervention, compared to placebo.

**Figure 2 brainsci-15-00761-f002:**
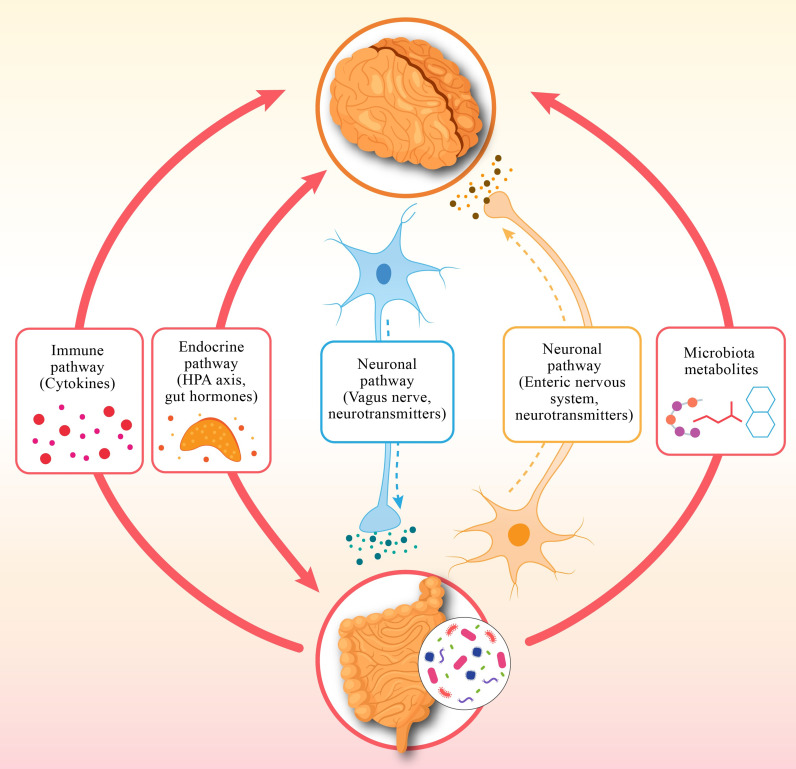
Figure displaying distributions of microbial metabolites (butyrate, propionate, kynurenine, indole-3-acetic acid) in participants with depressive symptoms, anxiety symptoms, and healthy controls.

**Figure 3 brainsci-15-00761-f003:**
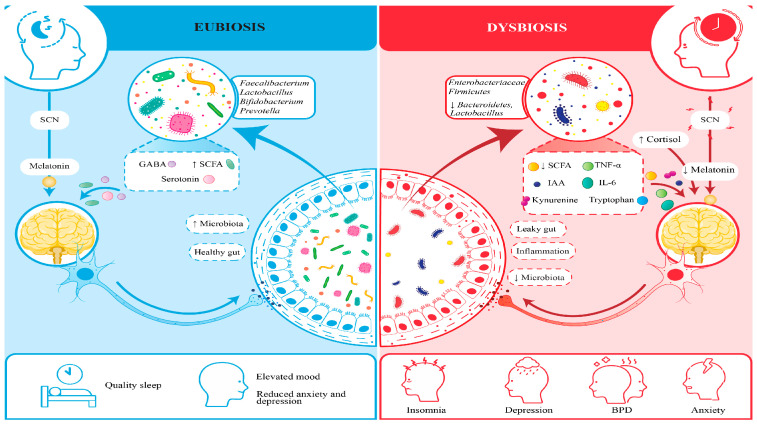
Figure comparing between participants with gut dysbiosis and healthy microbiomes.

**Figure 4 brainsci-15-00761-f004:**
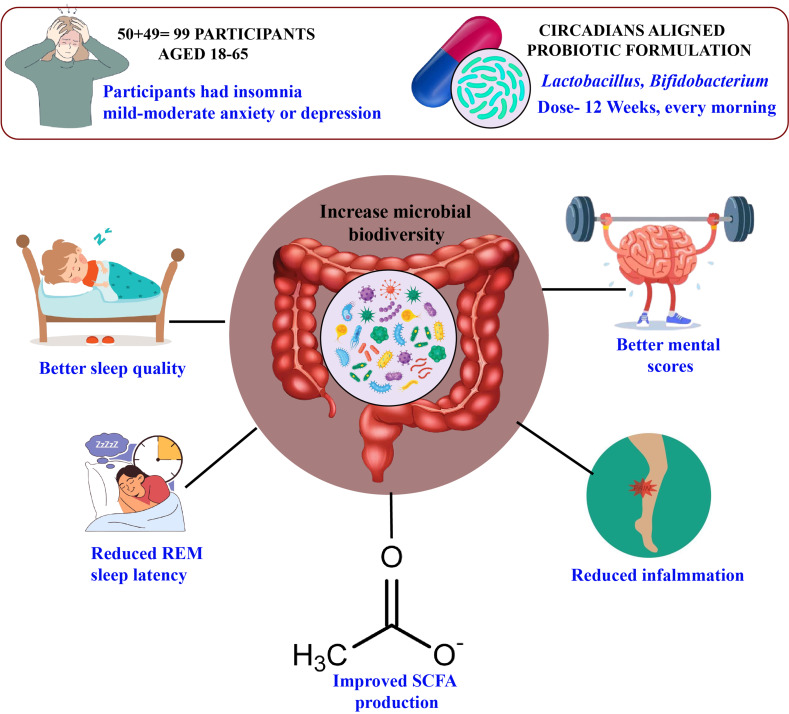
Figure showing relative abundances of bacterial taxa (e.g., Bacteroidetes, Firmicutes, *Lactobacillus, Bifidobacterium*) in participants with circadian misalignment vs. normal circadian rhythms.

**Table 1 brainsci-15-00761-t001:** Summary of Interventions, Dosage, Duration, and Compliance.

Q	Participants (n)	Intervention Content	Dosage/Day	Duration	Compliance Checks
Probiotic	50	*L. rhamnosus* GG + *B. longum*	15B CFU	12 weeks	Capsule counts + weekly telehealth
Placebo	49	Maltodextrin (matched appearance)	N/A	12 weeks	Capsule counts + weekly telehealth

**Table 2 brainsci-15-00761-t002:** Gut Microbiome Composition in Participants with Circadian Misalignment vs. Normal Circadian Rhythm.

Taxonomic Level	Circadian Misalignment (%)	*p*-Value
Bacteroidetes	35.2 ± 4.8	0.003
Firmicutes	50.1 ± 6.0	0.012
Lactobacillus	1.8 ± 0.6	<0.001
Bifidobacterium	2.1 ± 0.5	<0.001
Prevotella	5.3 ± 1.4	0.021
Akkermansia	0.9 ± 0.4	0.045

*p*-Values represent statistical comparisons between the circadian misalignment group (data shown) and the normal circadian rhythm group.

**Table 3 brainsci-15-00761-t003:** Sleep Architecture Metrics in Participants with Gut Microbial Dysbiosis vs. Healthy Microbiome.

Sleep Metric	Dysbiosis Group (Mean ± SD)	Healthy Microbiome Group (Mean ± SD)	*p*-Value
Sleep Efficiency (%)	78.3 ± 7.1	85.7 ± 6.6	<0.001
REM Sleep Latency (min)	120.5 ± 17.8	90.4 ± 14.9	<0.001
REM Sleep Duration (min)	70.2 ± 9.4	85.6 ± 9.1	<0.001
WASO (min)	45.6 ± 11.9	30.3 ± 10.2	0.002
N3 Sleep Duration (min)	80.5 ± 10.8	85.2 ± 10.2	0.045
Total Sleep Time (h)	6.5 ± 0.8	7.2 ± 0.7	0.004
Sleep Onset Latency (min)	25.3 ± 6.2	18.7 ±5.5	0.012

**Table 4 brainsci-15-00761-t004:** Microbial Metabolites Associated with Mental Health Symptoms.

Metabolite	Depressive Symptoms (Mean ± SD)	Anxiety Symptoms (Mean ± SD)	Healthy Controls (Mean ± SD)	*p*-Value
Butyrate (µg/mL)	12.3 ± 2.5	14.5 ± 2.7	18.7 ± 2.9	<0.001
Propionate (µg/mL)	8.5 ± 2.0	9.2 ± 2.2	12.4 ± 2.4	0.003
Kynurenine (ng/mL)	45.6 ± 6.0	55.3 ± 7.0	35.4 ± 5.5	<0.001
Indole-3-acetic acid (ng/mL)	22.3 ± 3.6	28.7 ± 4.0	18.5 ± 3.3	0.001

**Table 5 brainsci-15-00761-t005:** Changes in Sleep and Mental Health Metrics After Probiotic Intervention.

Metric	Probiotic Baseline (Mean ± SD)	Probiotic Post-Intervention (Mean ± SD)	Placebo Baseline (Mean ± SD)	Placebo Post-Intervention Mean ± SD)	*p*-Value *
Sleep Efficiency (%)	78.5 ± 7.0	86.2 ± 6.5	78.7 ± 6.9	79.3 ± 6.8	<0.001
REM Sleep Latency (min)	118.4 ± 16.0	92.3 ± 14.4	118.0 ± 15.8	115.6 ± 15.5	<0.001
Depression Score (PHQ-9)	12.3 ± 3.7	7.8 ± 3.0	12.1 ± 3.6	11.5 ± 3.6	<0.001
Anxiety Score (GAD-7)	10.5 ± 3.2	6.2 ± 2.5	10.3 ± 3.1	9.8 ± 3.1	<0.001

* Notes: *p-*values reflect comparisons between probiotic post-intervention and placebo post-intervention (independent *t*-tests). Baseline equivalence confirmed (probiotic vs. placebo): sleep efficiency (*p* = 0.887), REM sleep latency (*p* = 0.901), PHQ-9 (*p* = 0.786), GAD-7 (*p* = 0.753). Sample sizes: probiotic (*n* = 50), placebo (*n* = 49).

**Table 6 brainsci-15-00761-t006:** Post-Intervention Microbiome Changes.

Metric	Probiotic Baseline (Mean ± SD)	Probiotic Post-Intervention (Mean ± SD)	Placebo Post-Intervention (Mean ± SD)	*p*-Value *
Shannon Index	3.8 ± 0.6	4.2 ± 0.5	3.9 ± 0.6	0.002
Lactobacillus (%)	1.8 ± 0.6	3.5 ± 0.8	1.9 ± 0.7	<0.001
Bifidobacterium (%)	2.1 ± 0.5	4.8 ± 0.9	2.2 ± 0.6	<0.001
Butyrate (µg/mL)	12.3 ± 2.5	16.5 ± 2.7	12.5 ± 2.6	<0.001
Propionate (µg/mL)	8.5 ± 2.0	10.8 ± 2.1	10.8 ± 2.1	0.005

* Note: *p*-values reflect comparisons between probiotic post-intervention and placebo post-intervention (*t*-tests).

**Table 7 brainsci-15-00761-t007:** Pathway Analysis of Gut–Sleep–Brain Axis Interactions.

Pathway	Standardized Beta Coefficient	*p*-Value
Circadian Misalignment → Dysbiosis	0.45	0.005
Dysbiosis → Sleep Disturbances	0.38	0.012
Sleep Disturbances → Mental Health	0.52	0.002
Probiotic Intervention → Circadian Alignment	0.41	0.015
Circadian Alignment → Microbial Balance	0.37	0.020

**Table 8 brainsci-15-00761-t008:** Predictive Value of Microbial Metabolites for Mental Health Outcome.

Metabolite	AUC (Depression)	AUC (Anxiety)	*p*-Value
Butyrate	0.82	0.75	0.005
Kynurenine	0.78	0.84	0.007
Indole-3-acetic acid	0.71	0.69	0.025

## Data Availability

The original contributions presented in this study are included in the article. Further inquiries can be directed to the corresponding authors.

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
