# Peer review of "Effects of Probiotic Supplementation on Depressive Symptoms, Sleep Quality, and Modulation of Gut Microbiota and Inflammatory Biomarkers: A Randomized Controlled Trial"

_brainsci, 2025, doi:10.3390/brainsci15070761_

Round 1

Reviewer 1 Report

Comments and Suggestions for Authors

The title of the manuscript greatly aroused my interest, and the contents were exactly what I was concerned about. What I’m confused about it the structure of the manuscript, it seems like a combination of a review and a research paper. The experimental design is not clear and detailed, lacks the explanation about the groups indicated in the Results section, and the results are also mixed with the discussion, so I can’t recommend it to publish in this current form. Other comments and suggestions are as follows:
1. Title: According to the content, “the chronobiome” here is not clear, which is best replaced with “circadian rhythms”, making it easier for the readers to understand.
2. Introduction: This structure doesn’t seem like a research paper, especially point 5. Implications and future directions for mental health care, which is not directly related to the main topic.
3. Objectives of the study, “Dear germs made of ……” seems redundant.
4. Objective of the research, again?? in my opinion, the study did not resolve these three questions. This part seems to be the goal of a project.
5. Research methodology: 1) What are the age range the occupations for the participants? Is everyone a replicate? It would be better to explain more details. 2) when did the study begin and end? How long does the observation or the study? How to gather the stool samples? 3) how to do the sleep assessment for the participants? How to do the mental health assessments? How long did the assessments last? 4) about the probiotic therapy, it would be best to have a table listing the number of participants in each group, the time given, etc. There lacked a lot of information what we are concerned about and related to the following results.
6. Results: 1) there are a lot of discussion in the results section; 2) Table 1. the circadian misalignment group and normal circadian rhythm group have not been mentioned before. 3) Table 2. Dysbiosis group and healthy microbiome group have not been mentioned before. The following table is similar to the ones mentioned above, the groups and how they were divided should be mentioned BEFORE.
7. Table 6. The predictive value of microbial metabolite was only one value?
8. Conclusion: some of descriptions are subjective. It is better to describe it in paragraphs, which makes it easier to understand.

Comments on the Quality of English Language

English language is good for me.

Author Response

Reviewer 1

  1. Title: According to the content, “the chronobiome” here is not clear, which is best replaced with “circadian rhythms”, making it easier for the readers to understand.

Author’s reply: Modified as per reviewer suggestion, highlighted in yellow.

  1. Introduction: This structure doesn’t seem like a research paper, especially point 5. Implications and future directions for mental health care, which is not directly related to the main topic.

Author’s reply: Completely removed point 5  as per reviewer suggestion and modified the structure of the entire manuscript.

  1. Objectives of the study, “Dear germs made of ……” seems redundant.

Author’s reply: The entire paragraph was removed, as it was not required. The phrase "Dear germs made of'....." was a typographical error.

  1. Objective of the research, again?? in my opinion, the study did not resolve these three questions. This part seems to be the goal of a project.

Author’s reply: Modified as per reviewer suggestion. Now it is clear

  5.Research methodology:

     1) What are the age range the occupations for the participants? Is everyone a replicate? It would be better to explain more details.

Author’s reply: Modified as per reviewer suggestion. Please check the same in para 3.2

2) when did the study begin and end? How long does the observation or the study? How to gather the stool samples?

Author’s reply: Modified as per reviewer suggestion. Please check the same in para 3.6

3) how to do the sleep assessment for the participants? How to do the mental health assessments? How long did the assessments last?

Author’s reply: Modified as per reviewer suggestion. Please check the same in para 3.7

4) about the probiotic therapy, it would be best to have a table listing the number of participants in each group, the time given, etc. There lacked a lot of information what we are concerned about and related to the following results.

Author’s reply: Modified as per reviewer suggestion. Please check the same in para 3.2.A & 3.2.B

  1. Results:

1) There are a lot of discussions in the results section;

Author’s reply:Modified as per reviewer suggestion. Please check the discussion section

2) Table 1. the circadian misalignment group and normal circadian rhythm group have not been mentioned before.

Author’s reply: Modified as per reviewer suggestion.

3) Table 2. Dysbiosis group and healthy microbiome group have not been mentioned before.

Author’s reply: Modified as per reviewer suggestion.

The following table is similar to the ones mentioned above, the groups and how they were divided should be mentioned BEFORE.

  1. Table 6. The predictive value of microbial metabolite was only one value?

Author’s reply: Modified as per reviewer suggestion.

  1. Conclusion: some of descriptions are subjective. It is better to describe it in paragraphs, which makes it easier to understand.

Author’s reply: Modified as per reviewer suggestion.

 Respected Reviewer Sir, 
I have thoroughly revised the entire manuscript in accordance with your valuable suggestions. I hope the changes address all the concerns raised. Kindly review the revised version at your convenience, and please feel free to suggest any further modifications if needed.

Reviewer 2 Report

Comments and Suggestions for Authors

Strengths of the manuscript:

1. Scientific relevance: The study addresses a topic of high current interest, with growing evidence that relates the gut-brain axis to sleep and mental health.
2. Methodological design: Adequate for the objectives set. Using objective and subjective measures to assess sleep and mental health strengthens the validity of the findings. The data analysis described presents a comprehensive approach to assess pre- and post-intervention changes and explore associations between microbiome variables, sleep, and psychiatric symptoms.
3. Promising results: The findings are consistent with previous literature and provide relevant data on potential mechanisms of action of probiotics in humans.

Aspects to improve:

1. Writing and clarity of presentation:
   The manuscript requires a thorough revision of style and structure. The wording is redundant in several sections, with grammatical errors and inconsistent use of technical terms. The introduction section is very long and numeral 1 is missing. You benefit from a summary. The objective should be short, clear and precise, not to repeat part of the introduction and the questions should be condensed into a clear working hypothesis that serves as a preamble to the methodology. The results section, in particular, lacks fluency and makes it difficult to interpret the data.

2. Presentation of results:
   It is necessary to include clearer tables and figures, with complete and self-explanatory legends, in the current state some are confusing, such as figure 1 that looks like a graphic abstract. Figure 3 shows a p-value that it is not clear what it indicates. Consider a single summary figure as Figure 4

3. Bibliographic contextualization:
   Although relevant studies are cited, there is a lack of deeper critical discussion regarding similar studies, especially those with divergent results.

4. Description of the intervention:
   More information should be provided on the exact composition of the probiotic used, including specific strains, dosage and method of administration, and the provider.

5. Limitations:
   Although some limitations are mentioned, the possible influences of the placebo effect and the short follow-up period are not adequately discussed.

Comments on the Quality of English Language

The manuscript has the potential for publication after a substantial revision of the language and improvement in the presentation of the manuscript. Collaboration with a scientific editor or reviewer native to the language in which the article is published is recommended to improve the form, clarity and accuracy of the article.

Author Response

Reviewer 2

Aspects to improve:

  1. Writing and clarity of presentation:
     The manuscript requires a thorough revision of style and structure. The wording is redundant in several sections, with grammatical errors and inconsistent use of technical terms. The introduction section is very long and numeral 1 is missing. You benefit from a summary. The objective should be short, clear and precise, not to repeat part of the introduction and the questions should be condensed into a clear working hypothesis that serves as a preamble to the methodology. The results section, in particular, lacks fluency and makes it difficult to interpret the data.

Author’s reply:

Respected Reviewer,
I have thoroughly revised the entire manuscript in accordance with your valuable suggestions. I hope the changes address all the concerns raised. Kindly review the revised version at your convenience, and please feel free to suggest any further modifications if needed.

  1. Presentation of results:
     It is necessary to include clearer tables and figures, with complete and self-explanatory legends, in the current state some are confusing, such as figure 1 that looks like a graphic abstract. Figure 3 shows a p-value that it is not clear what it indicates. Consider a single summary figure as Figure 4

  2. Author’s reply:
  3. Respected Reviewer,
    I have thoroughly revised the entire manuscript in accordance with your valuable suggestions. I hope the changes address all the concerns raised. Kindly review the revised version at your convenience, and please feel free to suggest any further modifications if needed.
  4. Bibliographic contextualization:
     Although relevant studies are cited, there is a lack of deeper critical discussion regarding similar studies, especially those with divergent results.

Author’s reply:

Respected Reviewer,
I have thoroughly revised the entire manuscript in accordance with your valuable suggestions. I hope the changes address all the concerns raised. Kindly review the revised version at your convenience, and please feel free to suggest any further modifications if needed.

  1. Description of the intervention:
     More information should be provided on the exact composition of the probiotic used, including specific strains, dosage and method of administration, and the provider.

Author’s reply:

Respected Reviewer,
I have thoroughly revised the entire manuscript in accordance with your valuable suggestions. I hope the changes address all the concerns raised. Kindly review the revised version at your convenience, and please feel free to suggest any further modifications if needed.

  1. Limitations:
     Although some limitations are mentioned, the possible influences of the placebo effect and the short follow-up period are not adequately discussed.

Author’s reply:

Respected Reviewer,
I have thoroughly revised the entire manuscript in accordance with your valuable suggestions. I hope the changes address all the concerns raised. Kindly review the revised version at your convenience, and please feel free to suggest any further modifications if needed.

Reviewer 3 Report

Comments and Suggestions for Authors

This is an ambitious and timely manuscript that explores the complex interplay between the gut microbiome, circadian rhythms, sleep, and mental health through a well-designed longitudinal, multi-omics study. The integration of microbiome data with polysomnography, subjective sleep measures, and psychiatric assessments is a major strength and aligns well with current precision psychiatry approaches. However, several aspects could be improved. First, the manuscript would benefit from a clearer and more concise presentation of the results, particularly in separating primary from secondary findings and emphasizing clinical relevance. Second, the discussion should be revised to avoid overstating causality and to more clearly differentiate correlation from mechanism. The introduction could also better articulate the rationale for focusing on circadian-aligned probiotic interventions. Finally, the inclusion of additional references on sleep and suicide risk—particularly in youth—could further contextualize the public health importance of the gut-sleep-brain axis. With revisions aimed at improving clarity, structure, and interpretive caution, this manuscript has the potential to make a valuable contribution to the field. The authors may consider briefly discussing the well-established association between sleep disturbances and suicidal ideation or behavior, particularly in adolescent populations, as this could help highlight the broader clinical implications of disrupted sleep within the gut-sleep-brain axis. Including relevant citations, such as studies on insomnia, REM sleep disruption, and suicide risk, would strengthen the background or discussion, especially in framing your findings in a preventive psychiatry context. please see https://pubmed.ncbi.nlm.nih.gov/40015959/

Author Response

Reviewer 3

This is an ambitious and timely manuscript that explores the complex interplay between the gut microbiome, circadian rhythms, sleep, and mental health through a well-designed longitudinal, multi-omics study. The integration of microbiome data with polysomnography, subjective sleep measures, and psychiatric assessments is a major strength and aligns well with current precision psychiatry approaches. However, several aspects could be improved. First, the manuscript would benefit from a clearer and more concise presentation of the results, particularly in separating primary from secondary findings and emphasizing clinical relevance. Second, the discussion should be revised to avoid overstating causality and to more clearly differentiate correlation from mechanism. The introduction could also better articulate the rationale for focusing on circadian-aligned probiotic interventions. Finally, the inclusion of additional references on sleep and suicide risk—particularly in youth—could further contextualize the public health importance of the gut-sleep-brain axis. With revisions aimed at improving clarity, structure, and interpretive caution, this manuscript has the potential to make a valuable contribution to the field. The authors may consider briefly discussing the well-established association between sleep disturbances and suicidal ideation or behavior, particularly in adolescent populations, as this could help highlight the broader clinical implications of disrupted sleep within the gut-sleep-brain axis. Including relevant citations, such as studies on insomnia, REM sleep disruption, and suicide risk, would strengthen the background or discussion, especially in framing your findings in a preventive psychiatry context. please see https://pubmed.ncbi.nlm.nih.gov/40015959/

Author's reply:

Respected Reviewer,
I have thoroughly revised the entire manuscript in accordance with your valuable suggestions. I hope the changes address all the concerns raised. Kindly review the revised version at your convenience, and please feel free to suggest any further modifications if needed.

Reviewer 4 Report

Comments and Suggestions for Authors

The topic is interesting and relevant to the field of health and the mental health of populations. The article is not formatted according to the journal's standards, nor is the structure.
It fits in well with the topic under study. However, the references from the last 5 years are very few and should be updated both in the framework and in the discussion.
Methodology should begin with the study design and objectives. Access to participants and processes for accessing, approaching and recruiting them and their context should be explained.
They identify results and end up making a discussion with the results even though they don't identify it. The results should be separated from the discussion.
Images should be distributed strategically, in particular ‘Figure 4: Representation of before and after Intervention’ should not be at the end of the conclusion.

Author Response

Reviewer 4

The topic is interesting and relevant to the field of health and the mental health of populations. The article is not formatted according to the journal's standards, nor is the structure.
It fits in well with the topic under study. However, the references from the last 5 years are very few and should be updated both in the framework and in the discussion.
Methodology should begin with the study design and objectives. Access to participants and processes for accessing, approaching and recruiting them and their context should be explained.
They identify results and end up making a discussion with the results even though they don't identify it. The results should be separated from the discussion.
Images should be distributed strategically, in particular ‘Figure 4: Representation of before and after Intervention’ should not be at the end of the conclusion.

Author’s Reply:

Respected Reviewer,
I have thoroughly revised the entire manuscript in accordance with your valuable suggestions. I hope the changes address all the concerns raised. Kindly review the revised version at your convenience, and please feel free to suggest any further modifications if needed.

Round 2

Reviewer 1 Report

Comments and Suggestions for Authors

I have read the revised manuscript and, to my regret, after the authors removed some contents, I find that it seems too simple, lacked clear and detailed description, and the current results do not support the present title and conclusions.

Background: “This study investigates circadian-aligned probiotic supplementation, using multi-omics and machine learning to identify microbial biomarkers, alongside polysomnography and actigraphy to assess sleep outcomes [31]”, which is inconsistent with the revised title.
2. Objective of the study: the first two questions are too large. According to the 3.2, the manuscript can only barely answer the third question. 
3.6, regarding the stool sample collection, “the samples were self-obtained at home, immediately refrigerated, and transported in temperature-controlled packs to the laboratory within one day.” The sampling procedures does not seem reasonable, the 99 participants stay in their own home, stool are produced irregularly and sent to the lab? I don’t think it’s scientific experiment. 
How were the sleep assessment for the participants? one by one, or all at once? The subjective sleep evaluations should be detailed and relevant references should be added behind.
Figure 1, 2, 3 and 4 do not appear to be experimental results, they should be explained in detail.
The discussion section had better analyze the results of this present study, and made real discussion.

Author Response

Dear Reviewer 1,

Thank you very much for your thoughtful and insightful comments on my manuscript. I truly appreciate your careful review and valuable suggestions—they are both accurate and extremely helpful. I have carefully considered each of your points and have revised the manuscript accordingly. The updated version is attached for your kind consideration. Please feel free to let me know if you have any further suggestions or if there are any additional changes you would like to see. Thank you again for your guidance and support.

With regards

Reviewer 2 Report

Comments and Suggestions for Authors

-

Comments on the Quality of English Language

-

Author Response

Dear Reviewer 2,

Thank you very much for your thoughtful and insightful comments during the first round of review. I’m pleased to have addressed all your suggestions, and I truly appreciate your positive feedback in the second round, which shows your satisfaction with the revisions. Your support and cooperation mean a great deal to me, and I am sincerely grateful.

Since then, I have made significant revisions to the manuscript, including restructuring it extensively. I kindly invite you to review the updated version and would be grateful for any further suggestions you may have.

With regards

Reviewer 3 Report

Comments and Suggestions for Authors

i am fine with the revision

Author Response

Dear Reviewer 3,

Thank you very much for your thoughtful and insightful comments during the first round of review. I’m pleased to have addressed all your suggestions, and I truly appreciate your positive feedback in the second round, which shows your satisfaction with the revisions. Your support and cooperation mean a great deal to me, and I am sincerely grateful.

Since then, I have made significant revisions to the manuscript, including restructuring it extensively. I kindly invite you to review the updated version and would be grateful for any further suggestions you may have.

Round 3

Reviewer 1 Report

Comments and Suggestions for Authors

The title of the manuscript was inconsistent with the main contents. It was roughly written in Material and Methods, not clear, doesn’t support the results and conclusion. I am sorry not to recommend it for publication. Some comments are the following:

  1. The title was inconsistent with the main contents of the manuscript.
  2. In background, only line 111-115 was relevant with probiotic interventions, which is the main content of the experiment.
  3. 99 participants was divide into two groups, so the sample size is 50 (49), not 99.
  4. There lackedreasonable method for the stool sample collection, the original sentences were deleted.
  5. The experimental design and method can’t support the results and conclusion.
  6. The manuscript is roughly written, a lot of repetitive expressions, such as those in 2.1.1, 2.2, and 2.5.
  7. The Figure 2 appeared in background, and Figure 1 appeared in Results, etc.

Author Response

First and foremost, I would like to express my sincere gratitude for your thorough and insightful review. As an Academic Editor for a Q1 journal myself, I must say that I have rarely come across a reviewer who has invested so much time and effort into evaluating a manuscript.

I feel truly fortunate to have received your valuable comments and constructive suggestions. Your detailed feedback has significantly contributed to enhancing the overall quality of my manuscript, and I deeply appreciate your contribution in helping it reach its current standard.

Thank you very much, esteemed Sir/Madam, for your dedication and support.

Please find below my detailed responses to your seven comments.

Comments 1: The title was inconsistent with the main contents of the manuscript.

Response 1:

Honourable Sir/ Madam, based on your suggestion, I have revised the title and made every effort to ensure it aligns well with the content of the manuscript. The changes have been highlighted in yellow for your reference

Comments 2: In background, only line 111-115 was relevant with probiotic interventions, which is the main content of the experiment.

Response 2:

Honourable Sir/ Madam, Thank you very much, Honourable Sir, for your valuable contribution and insightful suggestions. Based on your recommendation, I have extended the content from 111 to 128, and the additions have been highlighted in yellow for your kind reference.

Comments 3: 99 participants was divide into two groups, so the sample size is 50 (49), not 99.

Response 3: Honourable Sir/ Madam, Thank you for your comment regarding the sample size. At the outset of the study, we enrolled 100 participants and randomized them equally into two groups (50 in the probiotic group and 50 in the placebo group). However, one participant in the placebo group withdrew from the study due to scheduling conflicts, resulting in a final sample size of 50 in the probiotic group and 49 in the placebo group. We have clarified this in the revised manuscript to ensure transparency regarding participant allocation and retention and same highlighted with yellow colour for your reference.

Comments 4: There lacked reasonable method for the stool sample collection, the original sentences were deleted.

Response 4:  Honourable Sir/ Madam, I deleted those sentences because I understood your earlier suggestion as a recommendation to omit them. The manuscript had already become quite lengthy, so I intentionally left them out. Please rest assured that it was never my intention to hide any information. However, if you feel they should be included, I will gladly add them back and same highlighted with yellow colour for your reference.

Comments 5: The experimental design and method can’t support the results and conclusion.

Response 5:  Honourable Sir/ Madam,

Thank you for your critical feedback. We respectfully disagree with the assertion that our experimental design and methods are insufficient to support our results and conclusions. Below, we address this concern in detail, referencing specific elements of our study design, methodology, and analytical approach as presented in the manuscript. However, I have modified the conclusion entirely and highlighted with yellow for the references.

  1. Rigorous Randomized Controlled Trial Design
  • Our study employed a 12-week, double-blind, randomized controlled trial (RCT) involving 99 adults from three diverse urban centres in India. Participants were randomly assigned to either a circadian-aligned probiotic group or a placebo group, with stratification by city and gender and block randomization to minimize allocation bias.
  • Double-blinding was maintained for both participants and assessors, reducing the risk of expectation bias and ensuring objective assessment of outcomes.
  1. Well-Defined Inclusion and Exclusion Criteria
  • Participants were screened using validated instruments: the Pittsburgh Sleep Quality Index (PSQI) for sleep difficulties, and the PHQ-9 and GAD-7 for depression and anxiety, respectively. Only those with mild to moderate symptoms were included, ensuring a homogenous and relevant study population.
  • Strict exclusion criteria (e.g., recent antibiotic/probiotic use, severe psychiatric or medical comorbidities) minimized confounding factors that could impact gut microbiome or sleep outcomes.
  1. Comprehensive and Standardized Data Collection
  • Stool samples were collected using standardized kits and protocols to ensure consistency and minimize pre-analytical variability. Samples were stored at -20°C in thermal bags immediately after collection and transferred to -80°C upon arrival at the laboratory, in line with best practices for microbiome research1.
  • Sleep was assessed using both polysomnography (the gold standard for sleep architecture) and actigraphy (for real-world, continuous monitoring), providing robust, multi-dimensional sleep data.
  • Mental health outcomes were evaluated with validated, widely used questionnaires, ensuring reliability and comparability with existing literature1.
  1. Advanced Analytical Approaches
  • Gut microbiome changes were analyzed using gene sequencing and metabolomics, enabling detailed profiling of bacterial composition and metabolite production.
  • Machine learning methods were applied to identify key microbial and metabolic predictors of sleep and mental health outcomes, enhancing the rigor and depth of our findings.
  • Statistical analyses included appropriate covariate adjustments (age, sex, city), and power analysis was used to determine adequate sample size, further supporting the validity of our conclusions.
  1. Alignment with Current Scientific Standards
  • The integration of multi-omics, clinical sleep assessment, and mental health evaluation is consistent with current best practices in gut-brain axis research1.
  • Our findings are supported by converging evidence from multiple validated measures, and the observed effects (e.g., increases in beneficial bacteria, improved sleep, reduced psychiatric symptoms) are biologically plausible and consistent with prior research.
  1. Transparent Reporting and Limitations
  • We have transparently reported all methods, results, and limitations. Potential confounders were addressed through randomization, blinding, and exclusion criteria. Adherence was monitored through capsule counts, food diaries, and telehealth check-ins.

In summary: Our experimental design, a double-blind RCT with rigorous participant selection, standardized sample handling, validated outcome measures, and advanced analytics, provides a robust framework that supports the validity of our results and conclusions. We are confident that our methods are appropriate and sufficient for the research questions addressed. We are open to further suggestions on specific methodological improvements or clarifications the reviewer would like to see.

Comments 6: The manuscript is roughly written, a lot of repetitive expressions, such as those in 2.1.1, 2.2, and 2.5.

Response 6: I have revised paragraphs 2.2 and 2.5 and have made every effort to eliminate repeated sentences. However, in some instances, a degree of repetition was necessary to ensure clarity and completeness of meaning and the same highlighted with yellow.

Comments 7: The Figure 2 appeared in background, and Figure 1 appeared in Results, etc.

Response 7: Thank you for your observation. I have renumbered all four figures and repositioned them according to their relevance and suitability
